# Quality Evaluation of Hainan Robusta Coffee Bean Oil Produced by Ultrasound Coupled with Coconut Oil Extraction

**DOI:** 10.3390/foods12112235

**Published:** 2023-06-01

**Authors:** Zheng Jia, Liting Wan, Zhaoxian Huang, Weimin Zhang

**Affiliations:** School of Food Science and Engineering, Engineering Research Center of Utilization of Tropical Polysaccharide Resources, Ministry of Education, Hainan University, Haikou 570228, China; 15037392180@163.com (Z.J.); litingwan@hotmail.com (L.W.); huangzhaoxian@hainanu.edu.cn (Z.H.)

**Keywords:** coconut oil, coffee beans, ultrasound treatment, quality evaluation

## Abstract

This study investigates the treatment of coconut oil using thermosonic treatment in combination with green coffee beans. Under a defined ratio of coconut oil to green coffee beans, the effect of different thermosonic time on the quality parameters, active substance content, antioxidant capacity, and thermal oxidative stability of coconut oil were investigated as a strategy to potentially improve the quality of oil. Results showed that the β-sitosterol content of CCO (coconut coffee oil) treated with the thermal method combined with green coffee bean treatment reached up to 393.80 ± 11.13 mg/kg without affecting the lipid structure. In addition, DPPH clearance equivalents increased from 5.31 ± 1.30 mg EGCG/g to 71.34 ± 0.98 mg EGCG/g, and the ABTS clearance equivalent was 45.38 ± 0.87 mg EGCG/g versus 0 for the untreated sample. The improvement in thermal oxidation stability of treated coconut oil is also significant. The TG (Thermogravimetry) onset temperature was elevated from 277.97 °C to 335.08 °C and the induction time was elevated up to 24.73 ± 0.41 h from 5.17 ± 0.21 h. Thermosonic treatment in combination with green coffee beans is an ideal option to improve the quality of coconut oil. The results of this article provide new ideas for the development of plant-blended oil products and the new utilization of coconut oil and coffee beans.

## 1. Introduction

Edible oil can be used as an alternative to organic solvents to extract fat-soluble active substances and oil from natural products without damaging the edibility of the extracted material [1]. Edible oil also has a higher extraction efficiency compared to organic solvents. On the other hand, the toxicity of edible oils is low and they have a limited impact on the environment, so edible oils are regarded as “green” solvents. The extraction of bioactive substances and oils from natural resources can improve the quality of vegetable oils, and has a wide range of applications. In recent years, these oils have been defined as “flavored oil”, “gourmet oil”, or “healthy oil”, and have attracted much attention by consumers [2]. Current methods for extraction of active substances in natural resources are organic solvent extraction, pressurized liquid extraction [3], hot water extraction [4], microwave-assisted extraction [5], ultrasound-assisted extraction [6], and supercritical CO_2_ extraction [7]. Li et al. [8] used different solvents to extract α-mangostin from mangosteen peel and found that coconut oil as an extractant could provide higher α-mangostin bio-accessibility and improve the extraction efficiency. He et al. [9] used seven different vegetable oils as an entrainer to extract docosahexaenoic acid (DHA) from *Schizochytrium limacinum*. The results revealed high extraction efficiency and high extraction yield when vegetable oils were used as an entrainer and this gives access to new oil products. Other studies also tried to use vegetable oils as co-solvents, e.g., the extraction of lycopene from tomatoes using hazelnut oil [10] and the extraction of carotenoids from carrot using canola oil [1]. However, no studies have been reported on the use of coconut oil to extract bioactive substances in coffee beans.

As a typical tropical woody oil crop, coconut oil contains 90% saturated fatty acids and a large amount of lauric acid, which is the only oil in edible oils that consists of medium-chain fatty acids, also known as lauric oil [11]. Medium-chain fatty acids (MCFA) are easily digested and absorbed by the body; therefore, they can provide rapid energy supply, improve metabolism, lower cholesterol levels [12], and have good oxidative stability [13]. In addition, coconut oil has been reported to be beneficial for weight loss due to its high MCFA content, and it can assist in controlling blood pressure and cholesterol levels, as well as be a potential food for Alzheimer’s disease treatment [14]. However, the content of bioactive substances in coconut oil was low, for example, its phytosterol content was 70 mg/100 g, while the content of tocopherol was only 4.2 mg/100 g [14]. Green coffee beans are rich in nutrients [15] and contain a large number of bioactive compounds, such as chlorogenic acid, caffeine, phytosterols, and trigonelline [16]. These components have various biological activities such as antihypertensive, anti-inflammatory, anti-viral, hypolipidemic, hypoglycemic, and neuroprotective [17]. In addition, the PUFA content reached 40% in green coffee oil [18]. The active substances in coffee are currently underutilized, and the used coffee still contains a large amount of oil and presents a large number of bioactive compounds that cannot be fully utilized [19]. Therefore, the combination of coconut oil with coffee beans by technological means can make up for the shortcomings of the low content of bioactive substances in coconut oil.

A potential new technology that may improve the extraction efficiency of lipophilic compounds from plants sources is ultrasound [20]. Ultrasound facilitates the release of extractable compounds by disrupting plant cell walls and enhancing mass transport of solvent from the continuous phase into plant cells [21]. In addition, the equipment is relatively inexpensive compared with other new alternative extraction techniques and shows high efficiency because of the mechanical effects [22]. Nevertheless, this technology also has some shortcomings. During ultrasound processing, the solvent cannot be renewed, and it cannot solve the problem of solvent evaporation [23]. However, using oil as a solvent can compensate for the disadvantages of ultrasound technology.

In the early stage, aiming at problems such as poor flavor, bitter taste, low quality, and the lack of a quality evaluation system of Hainan Robusta coffee beans, our research group took Hainan Robusta coffee beans and coconut oil as raw materials to explore the effects of different ultrasound treatments on the quality of Robusta coffee beans, and then developed coconut coffee products. Meanwhile, coconut coffee oil was obtained. However, the quality of the coconut coffee oil is unclear. To develop coconut coffee oil products, the optimal ultrasound processing conditions were determined previously by combining correlation analysis with statistical models. This work provides new ideas for the development of plant-blended oil products and the new utilization of coconut oil and coffee beans.

## 2. Materials and Methods

### 2.1. Materials

The coconut oil and Robusta coffee beans were provided by Ye Ze Fang Food Co., Ltd. (Baoting, China) and stored at 4 °C before use. Coconut oil was made by fresh coconut meat through cold pressing. Ripe Robusta coffee cherries were collected by hand during the main coffee harvest in early 2021. The “Fully Washed” method was used to treat coffee beans. All chemicals and solvents used in this study were of analytical reagent grade and were obtained from Sigma Co. (St. Louis, MO, USA).

### 2.2. Thermosonic Treatment of Coconut Oil and Green Coffee Beans

The coconut oil and green coffee beans were mixed in a beaker with a ratio of 5:8 (*wt:wt*) and placed in the ultrasonic processor device (XO-5200DTD, Atpiocn, Nanjing, China). Green coffee beans after ultrasound treatment still need to be roasted for further research and marketing, so green coffee beans were not ground when put together with coconut oil. The thermosonic treatment was performed under the conditions of 40% of the power, 70 kHz and 50 °C, and the thermosonic time was controlled for 15 min (CCO-1), 30 min (CCO-2), 45 min (CCO-3), 60 min (CCO-4), 75 min (CCO-5), and 90 min (CCO-6), respectively. Subsequently, samples were centrifuged at 4000 rpm and 30 °C for 5 min. The upper oil was collected and stored at 4 °C before analysis.

### 2.3. Color Measurement

The exterior color of CCOs was measured using a color meter (Spectrophotometer TC- PIIG, Beijing, China). The colorimetric spot diameter was 10 mm, and a white paper was used as the standard sample and expressed as L* (lightness), a* (red/green color), and b* (yellow/blue color) values.

### 2.4. Quality Indices

The acid value (AV), peroxide value (PV) and p-anisidine value (p-AV) of CCOs were estimated using Ca 5a-40, Cd 18–90, and Cd 8b-90 methods, respectively, from American Oil Chemists’ Society (American Oil Chemists’ Society, 2017; American Oil Chemists’ Society, 1997; American Oil Chemists’ Society, 2011).

AV: In short, 5 g of sample oil was added to alcohol in a 25 mL beaker. Then, the mixture was titrated with 0.1 N NaOH using phenolphthalein as an indicator.

PV: In brief, the oil sample (5.0 g) was dissolved in 50 mL of acetic acid-isooctane solution mixture. Then 0.5 mL of saturated KI solution was added and the mixture was shaken at least three times in 1 min before adding 30 mL of distilled water. Subsequently, the solution was titrated with 0.1 M sodium thiosulfate until the yellow iodine color disappeared. Finally, 0.1 M sodium thiosulfate was used to titrate the solution using starch as an indicator.

p-AV: The CCO sample (4.0 g) was dissolved in 50 mL n-hexane and the absorbance of this solution was measured at 350 nm via a spectrophotometer (754NPC, AUCY, China) with N-hexane as the control. Then 2 mL of 0.25% p-Anisidine in acetic acid (*w*/*v*) was added to 10 mL of the solution and stored in the dark for 10 min. One mL of p-Anisidine solution and 5 mL n-hexane were used to perform the control experiment.

### 2.5. Fatty Acid Composition

Fatty acid composition was determined by gas chromatography as reported by Cui et al. [24]. Fatty acids were separated on a 7890 A series Agilent system GC-FID with a CP-Sil 88 column (100 m × 0.25 mm 0.2 µm, Agilent, Santa Clara, CA, USA). Fatty acid methyl esters (FAMEs) were identified and quantified by comparing the retention times of FAMEs standards and calculating the relative peak areas, respectively. The standard mixture of fatty acid methyl esters (FAMEs, GLC-463) was purchased from NuChek−Prep (Elysian, SP, MN, USA).

### 2.6. Raman Spectroscopy

Raman spectra were acquired using a finder insight pro Raman spectrometer (ZOLIX/FI785E10W-Pro-MPCD, Zolix Analytical Instruments Co., Ltd., Beijing, China). A drop of CCO was placed on a slide for observation with the following parameters: an excitation wavelength of 785 nm, a laser power of 20%, a sweep range of 200–4000 cm^−1^, and a total of 20 scans per spectrum. The collected Raman spectra were analyzed by the INScan-Lite software (Zolix Analytical Instruments Co., Ltd., Beijing, China).

### 2.7. Phytosterol Content Determination

The content of phytosterols in CCO was determined according to a previously published method [25] with some modifications. The oil sample was saponified with ethanolic potassium hydroxide solution. Then phytosterol components were separated and identified by GC-MS on an Agilent 7890B/5977A system (Agilent, Santa Clara, CA, USA) with a SE-45 capillary column (50 m × 0.25 mm × 0.1 μm). The phytosterols were identified by their retention times and MS spectra.

The different types of phytosterols detected were fitted by the empirical model (Equation (1)) proposed by [26], which is widely used to fit the changes in bioactive substance content and antioxidant activity during the experiment [27].
X_t_ = X_0_ + (t/(K_1_ + K_2_t)) = t/(K_1_ + K_2_t)(1)
where t is time (min), X_0_ (g/L) and X_t_ (g/L) are the initial (assumed zero in all experiments, raw solvent) and measured phytosterols content at time t (min), respectively. K_1_ is Peleg’s rate constant (min gdb/mg), and K_2_ is Peleg’s capacity constant (gdb/mg).

### 2.8. DPPH/ABTS Radical Scavenging Assay

#### 2.8.1. DPPH Radical Scavenging Assay

The 10-times diluted sample solution (0.5 mL) was mixed with 3.5 mL of DPPH solution (0.2 mM) and reacted for 30 min in the dark, then the absorbance was measured at 517 nm using the UV spectrophotometer (U-T3, Yipu instrument manufacturing (Shanghai) Co., Ltd., Shanghai, China). The epigallocatechin gallate (EGCG) was used as a standard solution and the DPPH radical scavenging ability was established as an EGCG equivalent (mg/g of oil).

#### 2.8.2. ABTS Radical Scavenging Assay

The ABTS solution was prepared as previously reported [28].

The 0.1 mL oil sample was mixed with 3.9 mL diluted ABTS solution and reacted for 6 min in the dark; subsequently, the absorbance was measured at 734 nm. EGCG was used as a standard solution and the ABTS radical scavenging ability was established as an EGCG equivalent (mg/g of oil).

### 2.9. Thermal Gravimetric (TG)

TG analysis was performed on the TG analyzer (STA6000, Perkin Elmer, WLM, MA, USA). The sample (2 mg) was placed in an alumina crucible and heated to 500 °C at 10 °C/min with a continuous pure air flow at 10 mL/min. The obtained thermal curves were processed and graphically represented using the dedicated Pyris software from PerkinElmer Instruments. The mass change and its change rate were determined by TG curve and derivative thermogravimetric analysis (DTG) curve, respectively. The onset temperature of the thermal decomposition was calculated from the respective TG curves and DTG curves.

### 2.10. Oxidative Induction Time (OIT)

The OIT of CCO was evaluated by the accelerated oxidation test using a Metrohm 892 Rancimat instrument (Metrohm Corporation, Herisau, Appenzell Ausserrhoden, Switzerland). Briefly, CCO (3 g) was heated at 150 ± 1.6 °C and conducted to 20 L/h under a purified air inflow. OIT (h) was obtained from the system software.

### 2.11. Volatile Components

The volatile components in CCO were determined as reported by Ma et al., 2022 [29]. Adsorbed volatile components were extracted by headspace-solid phase micro-extraction (HS-SPME) and then immediately analyzed by a gas chromatograph-mass selective detector (GC-MS) (GC: 7890B, Agilent, USA; MS: 5977, Agilent, Santa Clara, CA, USA). Samples were separated on HP-5MS and CP-WAX (both 50 m × 0.25 mm, 0.20 µm). The n-alkanes were used to calibrate the retention index (RI) and then each compound was confirmed by RI. Compounds were identified according to NIST 17 mass spectra libraries and RI information.

### 2.12. Statistical Analyses

All the measurements were performed at least in triplicates and the results were reported as average values ± standard deviation. The data was processed using GraphPad Prism 5.0 (San Diego, CA, USA) and the analysis of variance (ANOVA) was analyzed using the SPSS statistical software (version 21.0, IBM SPSS Inc., Chicago, IL, USA) to test the significance. For this purpose, the Duncan test with a significance level of 95% (*p* < 0.05) was adopted. The physicochemical indexes, compositions of phytosterols, and oxidation stability and the antioxidant activity of the CCOs were normalized and merged into one matrix and then subjected to principal component analysis, Circos map analysis, and cluster heat map analysis to explore the marker compounds that could differentiate the CCO samples obtained by different thermosonic treatment methods.

## 3. Results

### 3.1. Color

The appearance and L* (lightness), a* (red/green color), and b* (yellow/blue color) values of CCO are presented in Figure 1. As displayed in Figure 1a, the unprocessed CCO (CCO-0) was light and bright in color. After thermosonic treatment for different times, all CCO samples were yellowish and could be distinguished from CCO-0 with the naked eye. Thus, thermosonic treatment with coffee beans significantly changes the color of CCO; however, ultrasonic time has little effect on it. Figure 1b showed that CCO-0 exhibited L*, a*, and b* values of 44.33, 1.05, and −3.17, respectively. With increasing thermosonic time, L* and a* values tended to decrease, while b* values tended to increase, suggesting that the CCOs turn dark and yellow after ultrasonic treatment with Robusta coffee beans. The color of the edible oils relied on the pigments present in them. Color materials extracted from the Robusta coffee beans during ultrasonic treatment may be attributed to the color change of the CCOs. Besides, Robusta coffee beans are rich in carbohydrates and chlorogenic acid, which may form small molecules of yellow pigment when heated [30], causing the coconut oil to turn yellow. The change in color of the CCOs may also be related to carotenoids in coffee beans. carotenoids is one of the main fat-soluble pigments in coffee beans. During ultrasonic treatment, they may transfer to coconut oil, thereby contributing to the characteristic yellow color of the CCOs.

### 3.2. Quality Indices

AV, PV, and p-AV were used to indicate the oxidation extent of CCO after different ultrasonic times with Robusta coffee beans. AV represents the amount of free fatty acids present in an oil, while PV reflects the primary oxidation level by testing the content of peroxide and hydroperoxide [31]. In turn, p-AV determines the concentration of secondary oxidation products. As shown in Table 1, AV values significantly increased after thermosonic treatment (*p* < 0.05, except CCO-6), which could be attributed to the free fatty acids generated by the lipolysis of triglycerides during ultrasonic extraction. Although the AV increased, they were in the range of 0.65–0.86 mg/g. The changing pattern of PV is slightly different from that of AV. At the initial stage of treatment (0–30 min), thermosonic treatment did not affect the PV (*p* > 0.05), but when ultrasonic time increased from 45 min to 90 min, the PV augmented from 0.90 to 1.06 mmol/kg, likely owing to the degradation products of unsaturated fatty acids in the CCOs caused by thermosonic treatment (50 °C used in this study). Nevertheless, the PV of all processed samples was still lower than the standard allowed by CODEX (<15 mmol/kg). Similarly, all processed samples had significantly higher p-AV (*p* < 0.05) in relation to the raw sample (CCO-0). The above results were consistent with a recent study showing that extraction time had significant (*p* < 0.05) negative effects on the p-AV of cottonseed oil during thermosonic treatment [32].

Compared to the DPPH radical scavenging activity of untreated CCO (CCO-0, 5.31 mg EGCG/g), the corresponding values for sonicated samples were significantly increased to 71.34 mg EGCG/g (*p* < 0.05). Furthermore, the DPPH radical scavenging activity increased significantly with increasing thermosonic time. A similar changing trend was observed for the ABTS radical scavenging activity, which is probably because the bioactive substances contained in green coffee beans were extracted into CCO during the extraction process. This phenomenon was also reported in the extraction of carotenoids from waste using linseed oil as a solvent [33]. Since the ABTS assay is more sensitive than the DPPH assay in identifying antioxidant activity due to its faster kinetic responses, [34] the results of ABTS (vs. DPPH) radical scavenging activity were apparently higher.

### 3.3. Fatty Acid Composition

The fatty acid composition of the CCOs obtained under different thermosonic treatment conditions was analyzed by GC and the results are summarized in Table 2. For all samples, the saturated fatty acids (SFAs) accounted for 94.30–95.06%, in which lauric acid (C12:0) was the dominant fatty acid (51.04–53.13%), followed by myristic acid (C14:0, 18.74–19.43%) and palmitic acid (C16:0, 8.29–8.53%) This is consistent with previous reports [14]. Therefore, CCO is classified as medium-chain triglycerides (MCTs), which has health benefits, such as preventing obesity, reducing the risk of cardiovascular diseases, regulating blood sugar, and antibacterial effects [35]. As shown in Table 2, thermosonic treatment tended to increase the content of C8:0 and C10:0 in CCO, especially after ultrasonic treatment for 45 min. This is probably due to the hydrolysis of triglycerides during ultrasonic extraction, and it agreed well with the increased AV values of the processed CCOs. Though the concentration of individual UFAs was not statistically changed (*p* > 0.05) after thermosonic treatment, total unsaturated fatty acids (∑ UFA) was significantly decreased (*p* < 0.05, except CCO-5), resulting in a significantly higher ratio of ∑ SFA to ∑ UFA (*p* < 0.05). Overall, thermosonic treatment led to significant increase of C8:0 and C10:0, and the loss of total UFAs. The emerging fatty acid arachidonic acid in CCOs was a characteristic fatty acid in green coffee beans.

### 3.4. Phytosterols

Phytosterols are the main oil-soluble bioactive substances in coffee beans and β-sitosterol is the main phytosterol in coffee beans [18]. As shown in Figure 2a,b, the untreated CCO (CCO-1) has a low concentration of β-sitosterol (64.24 mg/kg ± 2.85) and cholesterol (1.88 mg/kg ± 0.01), presumably due to the oil process. Longer thermosonic times favored the incorporation of β-sitosterol in the CCO. The β-sitosterol content increased significantly as the thermosonic time increased from 15 min to 75 min. Meantime, we found that the total phytosterols content of the coffee beans decreased, indicating that the phytosterols from the coffee beans entered the CCO. However, as the thermosonic time is further prolonged, the phytosterols solubility reaches its maximum, and some of the phytosterols may begin to be destroyed due to the associated high temperatures and lengthy thermosonic times, which could lead to a drop in phytosterol content levels. The β-sitosterol content was the highest after 75 min of ultrasonic treatment (CCO-5), reaching 393.80 mg/kg ± 11.14, which was significantly higher than the unprocessed CCO-0 (64.24 mg/kg ± 2.85). The change pattern of cholesterol resembled that of β-sitosterol. As a result of thermosonic treatment, the cholesterol content in CCO was significantly enhanced (73.28 mg/kg ± 1.45) in a time-dependent manner. Fitting β-sitosterol and cholesterol content with ultrasonic time using Peleg’s model (Figure 3c) yielded satisfactory results (both R2 = 0.92), suggesting that Peleg’s model was reliable enough to predict the phytosterols content of CCO. These results indicated that thermosonic treatment is conducive to the release of phytosterols, and overall, they increased with the prolongation of treatment time. During ultrasonic extraction, the collapse of cavitation bubbles breaks the barriers between phases, leading to emulsification by thermosonic jets that allow a liquid to reach the other solution [36]. The oil-soluble bioactive substance in the green coffee beans can then enter the CCO with the green coffee bean oil. The solubility of phytosterols in edible oils is dependent on the density, viscosity, and polarity of edible oils. Generally, the high viscosity of edible oils has been regarded as a major reason for the low extraction yield [37]. The high viscosity of edible oils prevents the diffusion of phytosterols into the oils and leads to reduced content. CCO is a SCT (Short-Chain Triglyceride) and MCT (Medium-Chain Triglyceride) rich oil.

Oil at high temperatures tends to cause loss of some natural bioactive ingredients like β-sitosterol [38]. One of the most important natural sources of dietary plant sterols is vegetable oils. Thus, the above results suggest that thermosonic treatment with green coffee beans can replenish the deficiency of phytosterols in CCO. We also determined the total phenol content, tocopherol content, and composition of CCOs, but none was detected.

### 3.5. Oxidation Stability

OSI indicates the development of lipid oxidative products and the deterioration of oil during storage, heating, and frying [39], and depends on the amount of antioxidants in the oils [40]. The OSI of untreated CCO-0 was 5.17 ± 0.21 h. As shown in Figure 3, and the OSI increased with time during thermosonic treatment. The OSI nearly reached the maximum of 24.73 ± 0.41 h at thermosonic treatment for 90 min, which was nearly fivefold as much as that of CCO-0. Thermosonic treatment significantly increased the OSI of CCOs (*p* < 0.05). The green coffee oil (GCO) was extracted from green coffee beans considered a natural source of several bioactive compounds with desirable biological properties [41]. This result may be related to the synergistic antioxidant effect of lipid nutrients (phytosterols) extracted from coffee beans by thermosonic treatment, thereby increasing the OSI of CCOs.

The thermal stability of all sample oil was determined by TGA analysis and the results of the TG and DTG plots are represented in Figure 3. The TG curve represents the thermal degradation of a compound. The onset temperature for the CCO-0, CCO-1, CCO-2, CCO-3, CCO-4, CCO-5, and CCO-6 were noted at 277.9 °C, 290.61 °C, 300.65 °C, 302.50 °C, 304.67 °C, 322.42 °C, and 335.08 °C, respectively. The high starting temperature of the oil sample indicates that the antioxidant components in the oil inhibit thermal oxidation. The TG was consistent with the OSI results. The main thermal decomposition of oil samples occurred between 277.97 °C and 335.08 °C, and a sharp decline in the weight percentage of each oil sample was observed, suggesting decomposition of monounsaturated fatty acids, saturated fatty acids, and different hydro-peroxides present in the oil. The content of β-sitosterol in other samples was significantly higher than that in CCO-0. Phytosterols can delay lipid oxidation, but they may be degraded in this process [42]. Winkler et al. [43] discovered that the induction period of soybean and sunflower seed oils could be reduced by the addition of phytosterols (e.g., stigmasterol and sitosterol), and Dutta [44] found that adding β-sitosterol could improve sunflower oil’s performance when frying. The ethyl of sitosterol is attached to the R chain, which is more stable than other phytosterols and contributes to its antioxidant activity [45]. However, the antioxidant mechanism of phytosterols is still unexplored.

### 3.6. Raman Spectra

Figure 4 showed the Raman spectra of CCOs after thermosonic treatment for different times. The Raman peaks of all oil samples were mainly concentrated in the range of 800–1800 cm^−1^ with characteristic vibration patterns at 872, 1080, 1315, 1380, 1453, and 1755 cm^−1^. The peak of about 872 cm^−1^ indicates saturated fatty acids. The peak at about 1080 cm^−1^ shows the (Csingle bondC) stretching vibration of the (CH_2_)n group. The signal at 1315 cm^−1^ is designated as the (Csingle bondH) bending twist of the CH_2_ group. The signal at 1380 cm^−1^ can be assigned to the (Csingle bondH) symmetrical bending of CH_2_, the signal at 1453 cm^−1^ can be assigned to the (Csingle bondH) scissoring of CH_2_, and the mode at 1755 cm-1 indicated the (Cdouble-bondO) telescopic vibration of RCdouble bondOOR.

In short, Raman spectroscopy showed that the thermosonic treatment did not affect the structure and composition of the coconut oil, but maintained its high MCFA content.

### 3.7. Volatile Component

Aroma significantly affects quality and is the key criterion for consumers when choosing food [46]. Volatile components in CCO were determined by HP-SPME-GC-MS. Among them, as shown in Figure 5b, 11 VOCs were detected in CCOs, including 4 esters, 2 ketones, 2 hydrocarbons, 1 alcohol, 1 acid, and 1 organic urea compound. Diazene, dimethyl- was detected only in CCO-0 and CCO-1, and 4-methylhydrazinecarbothioamide was detected only in CCO-5 and CCO-6. PCA was used to analyze the VOCs listed in Figure 5a. CCO-0 was in the first quadrant alone, CCO-5 and CCO-6 was in the second quadrant. Other samples were located on the underside (negative side) of PC2, the result of PCA showed that the difference between CCOs was substantial.

δ-octalactone, ∑-nonalactone, and δ-hexalactone were the main volatile components in untreated CCO (CCO-1), accounting for ~80%. Consistently, previous research reported the saturated δ-lactones (e.g., δ-octalactone and ∑-nonalactone) being the most abundant volatile components in coconut oil [47], which contributed to the sweet flavor of coconut oil. With the increase of thermosonic time, the relative content of butanediol, which provides creamy sweet smell [48], increased first and then decreased. 2-heptanone has a fruit flavor similar to pear [49], and its relative content was negatively correlated with treatment time. The relative content of acetic acid was positively correlated with the treatment time, possibly due to the oxidative decomposition of coconut oil under thermosonic conditions. Sucrose is reported to be a precursor of fatty acids such as acetic acid [50]. Therefore, the elevated acetic acid content may also be due to the degradation of sucrose in green coffee beans during ultrasonic extraction. 4-Methylthiosemicarbazide was only detected in the CCO-5 and CCO-6, which were sonicated for a long time (75 min and 90 min, respectively). It has a weak ammonia odor, which may be related to the decomposition of protein in Robusta coffee beans during thermosonic treatment. Nevertheless, its relative content was low (0.15%, CCO-5; 0.14%, CCO-6). In general, adding green coffee beans combined with thermal thermosonic treatment did not significantly change the main flavor substances of coconut oil (δ-octalactone, ∑-nonalactone, and δ-hexalactone), which is also the desired result.

### 3.8. Relevance Analysis

All parameters were normalized to provide equal contributions of the variables in the predicted outcome before chemometric processing. PCA was performed on the fused matrix composed of 7 rows corresponding to the samples, and 9 columns that were composed of the physicochemical indexes, antioxidant activity index, β-sitosterol, cholesterol, oxidative stability index, and the antioxidant activity of sample oil. As shown in Figure 6a, the first principal component (PC1) explained 85.4% of the total variance, the second principal component (PC2) explained 10.7% of the total variance, and the score plot expresses a clear distribution of the oils according to the indexes. Figure 6b showed the loading plot in the PC1-PC2 plane. PC1 was positively related to any parameters. The β-sitosterol exhibited the highest positive loadings on PC1, from which we can affirm that the CCO sample was characterized by its high level of β-sitosterol. The extraction methods can produce CCOs with different characteristics, which could define their special use in the food industry.

Circos maps have been widely used in comparative genetics and foods, and help to visualise relationships between individuals with multidimensional data [51]. The samples in this study were divided into 7 groups according to the thermosonic time (Figure 7b), distributed from right to left in the down region. For cholesterol, OSI value, TG value, β-sitosterol, DPPH, and ABTS, it can be observed that the connecting line representing these indicators gradually thickens from left to right, and is the highest in CCO-6; this indicates that CCO-6 had the highest content of active substances, the best antioxidant activity, and oxidative stability, and that the addition of coffee beans combined with ultrasonic treatment could effectively enhance the functionality of coconut oil.

The heat map can reflect the index differences of samples prepared by different pretreatment methods. In the heat map (Figure 7a) composed of 7 groups of samples and 9 groups of indicators, red and blue indicate high and low levels, respectively. At the same time, 7 groups of samples were clustered and grouped by the heat map. The untreated sample CCO-0 was separately classified into one class, CCO-1 and CCO-2 were classified into one class, CCO-3 and CCO-4 were classified into one class, and CCO-5 and CCO-6 were classified into one class; these clustering trends were in accordance with those in the PCA score bioplot (Figure 6a) and Circos map (Figure 7b). According to the results of cluster analysis and Circos map, clustering demonstrated that CCO-5 and CCO-6 have higher content, β-sitosterol has stronger DPPH, and ABTS scavenging capacity and better antioxidant activity; therefore, it was separated from other samples. CCO-0, which contained the lowest β-sitosterol content, was divided into a separate class.

## 4. Discussion

In China, vegetable oils are often used for high-temperature cooking or frying, so it is necessary to find appropriate ways to improve the antioxidant capacity and oxidation stability of vegetable oils. The quality evaluation of coconut coffee oil in this work was mainly conducted from four aspects: physical and chemical properties, active substances, antioxidant capacity, and oxidation stability.

During ultrasonic treatment of CCOs, the mixture of oxygen and water was limited, and the oxidation of oil was difficult to conduct. However, there were cavitation and mechanical effects during ultrasound processing. When the bubbles collapse, the extreme environment with high temperature and pressure may cause the vegetable oil to oxidize faster. With the increase of ultrasound time, the AV, PV, and p-AV of CCOs showed small increases, but they were still within the safe range (AV < 4.0 mg KOH/g Oil; PV < 15 milliequivalents of active oxygen/kg oil). The thermal oxidation of oil is mainly due to the formation of small molecules from aldehydes and ketones once the double bonds of UFAs are broken. Coconut oil, on the other hand, has a high content of SFAs, and is thus stable. As a result, ultrasonic treatment had little effect on the fatty acid composition of coffee coconut oil, which was mainly composed of SFAs.

Ultrasound treatment can greatly improve the antioxidant capacity and oxidation stability of coffee coconut oil, and this effect was positively correlated with ultrasound time. In coffee beans, cells are encased in cell walls that act as a mechanical barrier during the flow of material out of the cell. There are also interactions between bioactive substances and other macromolecular substances. All of these hinder the mass transfer. The cavitation effect and mechanical effect produced during ultrasonic treatment can effectively destroy the cell wall, and accelerate the movement of molecules in the substance, so that the bioactive substance can quickly dissociate into the solvent. As shown in Figure 2, the content of phytosterols increased with the increase of ultrasonic time. The composition and content of tocopherol were also determined, but none was detected. This may be because the content of tocopherol in the sample is too low to reach the detection limit. Due to its triterpene structure, β-sitosterol is regarded as a natural antioxidant and has a moderate free radical scavenging ability. The increased phytosterol content in CCOs was the main reason for the improvement of their antioxidant capacity and thermal oxidation stability. Phytosterols are demonstrated to have various bioactive properties. For example, there were studies revealing the anti-isomerization mechanisms of phytosterols on trans fatty acids (TFAs) in peanut oil [52]. In this study, the microscopic morphology of coffee beans before and after ultrasonic treatment was also examined (Appendix A). Accordingly, it was found that micro-pores and partial protrusions appeared on the surface of coffee beans after ultrasonic treatment, indicating that material exchange occurs between coffee beans and coconut oil during ultrasonic treatment.

## 5. Conclusions

In the present study, a method for developing a coconut coffee oil product was established. Thermosonic treatment had a significant effect on the sterol content, antioxidant capacity, and thermal oxidative stability of the CCOs. Though the quality index of the sample oil increased, they were still within the safe range. Ultrasonic treatment showed no significant effect on the Raman spectra and the fatty acid composition of CCOs.

The thermal and cavitation effects of thermosonic treatment dissolved some of the bioactive substances in the green coffee beans into the coconut oil, which not only increased the bioactive substance content of the CCOs, but also enhanced their antioxidant capacity and thermal oxidation stability; moreover, increasing thermosonic treatment time promoted these changes. In this study, an oxidative stable coffee coconut oil with enhanced bioactive components was produced, which may be used as a potential cooking oil that benefits human health in the future.

## Figures and Tables

**Figure 1 foods-12-02235-f001:**
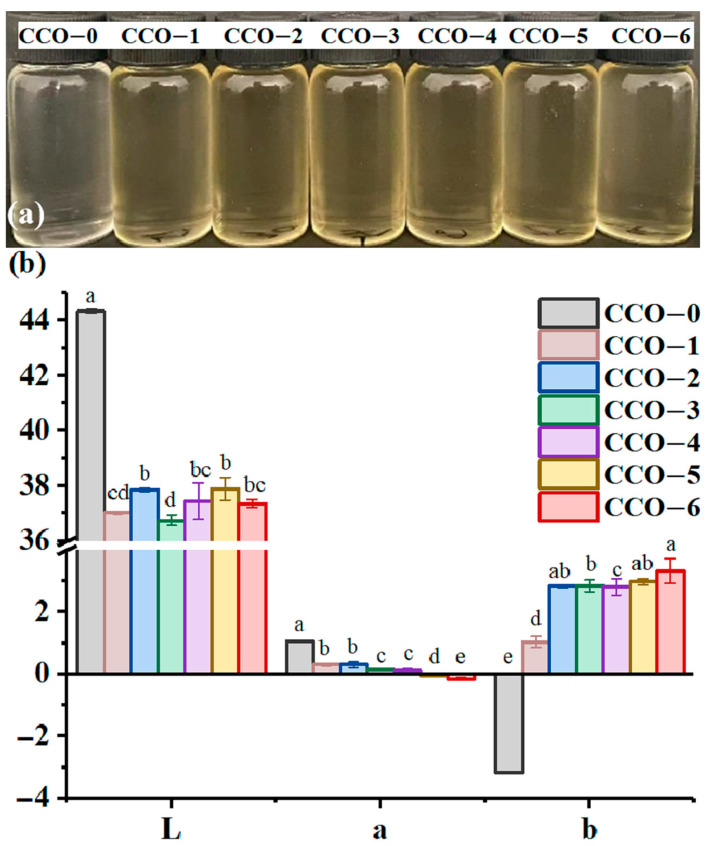
Lab value (**a**) and color change (**b**) of coconut oil during different thermosonic treatments. Different letters in the same row indicate statistically significant differences (*p* < 0.05).

**Figure 2 foods-12-02235-f002:**
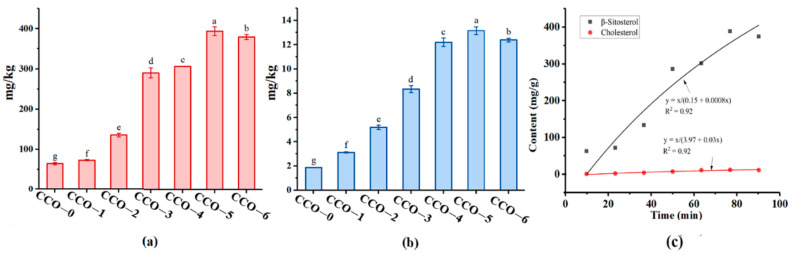
Phytosterol content of coconut oils during different thermosonic treatments (**a**); extraction kinetics of β-sitosterol (**b**); extraction kinetics of cholesterol (**c**); adsorption kinetics simulation. Different letters in the same row indicate statistically significant differences (*p* < 0.05).

**Figure 3 foods-12-02235-f003:**
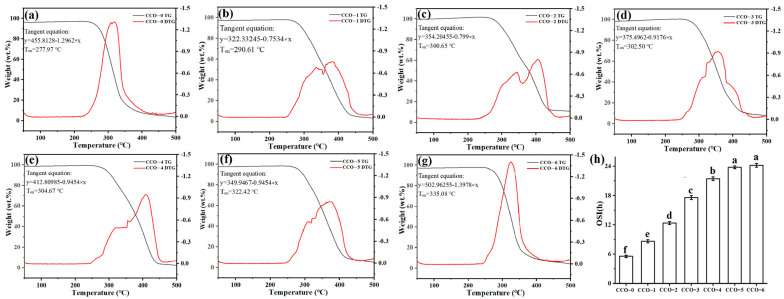
TG and OSI of coconut oils during different thermosonic treatments. (**a**) TG results of CCO-0, (**b**) TG results of CCO-1, (**c**) TG results of CCO-2, (**d**) TG results of CCO-3, (**e**) TG results of CCO-4, (**f**) TG results of CCO-5, (**g**) TG results of CCO-6, (**h**) OSI results of CCOs. Different letters in the same row indicate statistically significant differences (*p* < 0.05).

**Figure 4 foods-12-02235-f004:**
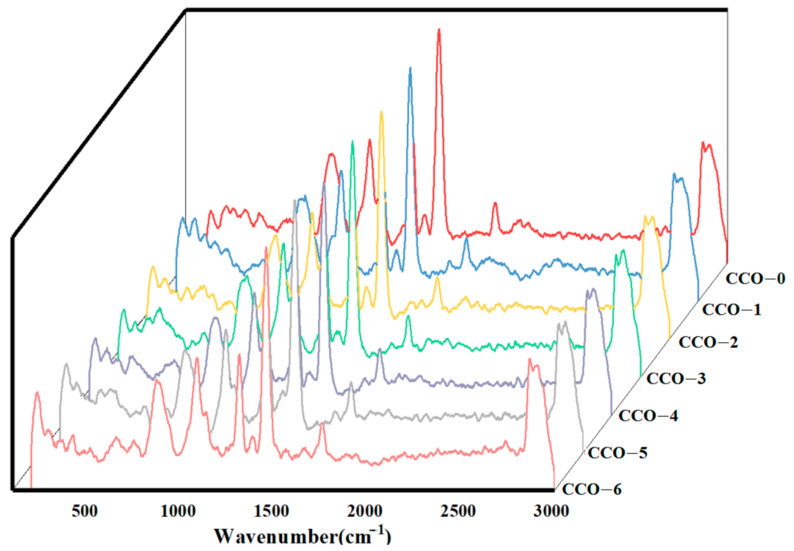
Raman spectra of coconut oils during different thermosonic treatments.

**Figure 5 foods-12-02235-f005:**
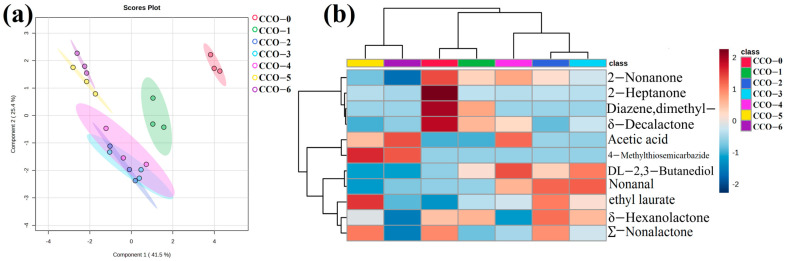
PCA of CCOs based on volatile compounds (**a**) cluster heat map hierarchically using the fused matrix based on volatile compounds (**b**) dataset matrix.

**Figure 6 foods-12-02235-f006:**
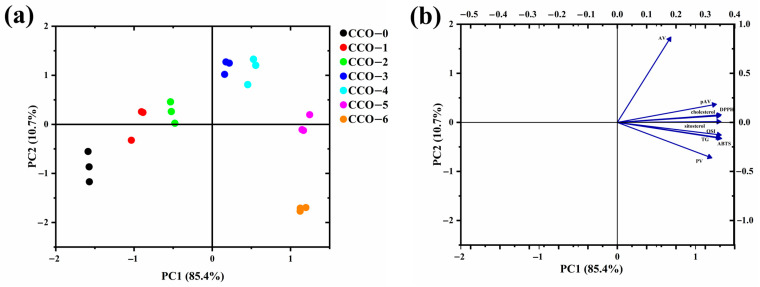
PCA bioplot obtained from the fused dataset matrix of CCOs from the different thermosonic treatments ((**a**): score biplot; (**b**): loading biplot).

**Figure 7 foods-12-02235-f007:**
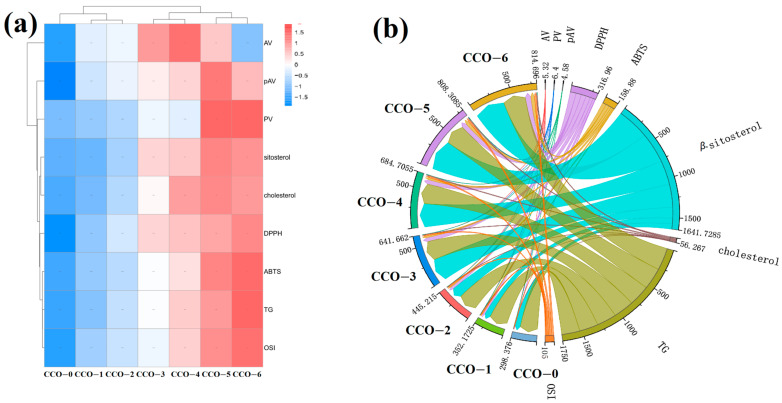
Circos map displaying the relationships among the oils, physicochemical indexes, phytosterols, and antioxidant activity and oxidation stability (**a**); cluster heat map hierarchically using the fused dataset matrix (**b**).

**Table 1 foods-12-02235-t001:** The quality indices of CCO during different thermosonic treatments.

Indices	CCO-0	CCO-1	CCO-2	CCO-3	CCO-4	CCO-5	CCO-6
AV ((KOH)/ (mg/g))	0.65 *±* 0.01 ^c^	0.74 *±* 0. 01 ^b^	0.75 *±* 0.02 ^b^	0.83 *±* 0.01 ^a^	0.86 *±* 0.02 ^a^	0.80 *±* 0.03 ^a^	0.69 *±* 0.01 ^c^
PV (mmol/kg)	0.81 *±* 0.01 ^c^	0.83 *±* 0.01 ^c^	0.85 *±* 0.01 ^bc^	0.90 *±* 0.02 ^b^	0.89 *±* 0.04 ^b^	1.06 *±* 0.03 ^a^	1.06 *±* 0.03 ^a^
p-AV (meq/kg)	0.49 *±* 0.01 ^d^	0.62 *±* 0.02 ^c^	0.64 *±* 0.02 ^bc^	0.67 *±* 0.03 ^b^	0.69 *±* 0.05 ^b^	0.76 *±* 0.03 ^a^	0.71 *±* 0.04 ^ab^
DPPH (mg EGCG/g)	5.31 *±* 1.30 ^g^	25.04 *±* 2.50 ^f^	36.85 *±* 1.61 ^e^	54.62 *±* 2.07 ^d^	58.64 *±* 1.37 ^c^	65.16 *±* 2.88 ^b^	71.34 *±* 0.98 ^a^
ABTS (mg EGCG/g)	0 *±* 0.00 ^g^	8.54 *±* 1.02 ^f^	13.25 *±* 1.74 ^e^	22.50 *±* 2.01 ^d^	27.64 *±* 0.48 ^c^	41.57 *±* 1.54 ^b^	45.38 *±* 0.87 ^a^

Different letters in the same row indicate statistically significant differences (*p* < 0.05).

**Table 2 foods-12-02235-t002:** Fatty acid composition (%) of coconut oil during different thermosonic treatments.

Fatty Acids	CCO-0	CCO-1	CCO-2	CCO-3	CCO-4	CCO-5	CCO-6
C8:0	3.82 *±* 0.55 ^b^	5.03 *±* 0.80 ^a^	4.97 *±* 0.97 ^ab^	5.28 *±* 0.62 ^a^	6.14 *±* 0.38 ^a^	6.23 *±* 0.28 ^a^	6.10 *±* 0.73 ^a^
C10:0	6.47 *±* 0.99 ^b^	6.67 *±* 0.16 ^ab^	6.57 *±* 0.07 ^ab^	6.58 *±* 0.22 ^ab^	6.76 *±* 0.04 ^a^	6.56 *±* 0.08 ^ab^	6.72 *±* 0.13 ^a^
C12:0	53.13 *±* 0.02 ^a^	52.28 *±* 0.14 ^ab^	52.23 *±* 0.10 ^ab^	51.72 *±* 0.57 ^ab^	51.81 *±* 1.97 ^ab^	51.04 *±* 0.75 ^b^	51.71 *±* 0.34 ^ab^
C14:0	19.43 *±* 0.97 ^a^	19.18 *±* 0.67 ^abc^	18.82 *±* 1.00 ^bc^	19.21 *±* 0.28 ^ab^	18.80 *±* 0.23 ^bc^	18.74 *±* 0.34 ^c^	18.85 *±* 0.14 ^bc^
C16:0	8.53 *±* 0.08 ^ab^	8.47 *±* 0.25 ^ab^	8.40 *±* 0.20 ^ab^	8.59 *±* 0.32 ^a^	8.29 *±* 0.20 ^b^	8.41 *±* 0.17 ^ab^	8.34 *±* 0.15 ^ab^
C18:0	3.22 *±* 0.18 ^a^	3.22 *±* 0.09 ^a^	3.23 *±* 0.14 ^a^	3.35 *±* 0.02 ^a^	3.23 *±* 0.16 ^a^	3.25 *±* 0.09 ^a^	3.26 *±* 0.10 ^a^
C18:1	4.41 *±* 0.05 ^a^	4.19 *±* 0.11 ^a^	4.30 *±* 0.08 ^a^	4.23 *±* 0.12 ^a^	3.95 *±* 0.08 ^a^	4.85 *±* 0.09 ^a^	4.09 *±* 0.12 ^a^
C18:2	0.70 *±* 0.07 ^a^	0.66 *±* 0.40 ^a^	0.71 *±* 0.10 ^a^	0.66 *±* 0.07 ^a^	0.62 *±* 1.52 ^a^	0.64 *±* 0.20 ^a^	0.64 *±* 0.58 ^a^
C20:0	0.00 *±* 0.00 ^a^	0.09 *±* 0.01 ^b^	0.08 *±* 0.13 ^b^	0.12 *±* 0.03 ^b^	0.11 *±* 0.01 ^b^	0.08 *±* 0.01 ^b^	0.08 *±* 0.06 ^b^
C20:1	0.29 *±* 0.03 ^a^	0.22 *±* 0.10 ^a^	0.17 *±* 0.04 ^a^	0.26 *±* 0.05 ^a^	0.28 *±* 0.01 ^a^	0.20 *±* 0.03 ^a^	0.22 *±* 0.09 ^a^
SFAs	94.6 *±* 0.98 ^a^	94.94 *±* 1.20 ^a^	94.3 *±* 0.74 ^a^	94.85 *±* 0.89 ^a^	95.14 *±* 1.04 ^a^	94.31 *±* 0.84 ^a^	95.06 *±* 0.95 ^a^
UFAs	5.4 *±* 0.14 ^b^	5.07 *±* 0.11 ^c^	5.18 *±* 0.20 ^b^	5.15 *±* 0.18 ^b^	4.85 *±* 0.09 ^d^	5.69 *±* 0.14 ^a^	4.95 *±* 0.45 ^d^
SFAs/UFAS UFAs AAAs	17.52	18.72	18.20	18.42	19.62	16.57	19.20

Different letters in the same row indicate statistically significant differences (*p* < 0.05).

## Data Availability

Data is contained within the article.

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
