# Peer review of "Quality Evaluation of Hainan Robusta Coffee Bean Oil Produced by Ultrasound Coupled with Coconut Oil Extraction"

_foods, 2023, doi:10.3390/foods12112235_

Round 1

Reviewer 1 Report

In an attempt to improve the quality of coconut oil and develop a new plant blended oil product – coconut cofee oil, coconut oil combined with green coffee beans were subjected to thermosonic treatment at different time intervals and the quality (physical and chemical properties, antioxidant capacity and oxidation stability) of obtained oli was evaluated. This paper is of high quality from a scientific aspect, it included a lot of analytical work with a clearly presented interpretation of the results, but some minor objections are as follows:

Line 13-19 It is not presented how much the initial values of the results for these parameters are, so the reader has no idea how much the antioxidant capacity, oxidative stability, etc. has improved. Present it differently to make it stand out.

Line 18, Introduce the full term for the abbreviation TG

Write keywords in lower case.

Line 29, The same sentence is repeated, throw it out.

Line 77-78 type of coffee- write in italics.

Line 87-90, Write more information about the origin and composition of coconut oil and coffee.

Throughout the text in some places it is not clear whether it is a mixture of coconut oil and coffee beans or coconut oil and cofee beans oil, pay attention to it.

2.3. and 2.4. Give more explicit information about these methods.

Line 108, as reported by.. Provide surname of the first author (and et al. if there are collaborators).

Line 192-194, Provide a reference for this assumption.

Line 198-202, Line 332, It is necessary to indicate a literary source for these sentences.

Line 229-244, compare these results with the results of related research in the literature.

Line 273, Full names for SCT and MCT.

Figure 1, Table 1, Table 2, Figure 2, missing: Different letters in the exponent in the same column of the table indicate a statistically significant difference between values, at a significance level of  p<0,05.

In conclusion CCOs, CCO and GCB, do not use abbreviated but full names, it should be clearer.

Author Response

Thank Reviewer for your reminder. Please see the attachment.

Reviewer 2 Report

This study describes the physicochemical characterization of a “coconut coffee oil” (CCO) which is produced by thermal ultrasonic treatment of coconut oil with coffee beans. The idea is to transfer bioactive compounds from the coffee to the coconut oil to obtain a CCO with enhanced properties. The aim of this study is clearly stated in the manuscript, but some modifications are needed before recommending publication.

According to the results, the larger the time of treatment, the higher the antioxidant capacity of CCO (measured by both DPPH and ABTS) and the content of B-sitosterol. However, authors only measured this bioactive compound. Coffee beans are source of other relevant compounds with antioxidant properties such as phenols, carotenoids, and tocopherols, as it has been described in previous published works. Release of these compounds from coffee beans to the coconut oil could contribute also to the higher antioxidant capacity and OSI of CCO in a relevant way. However, this issue has been unexplored by the authors. For instance, it has been reported that coffee beans contain chlorogenic acid and related compounds as main phenols. It would have been advisable that authors measured at least chlorogenic acid in coffee beans and in CCO. The same applies for carotenoids. Authors say that tocopherols were analyzed in CCO but not detected (lines 431 and 432). It raises the question of whether tocopherols were available in coffee beans and were not transferred to the coconut oil, or whether tocopherols were not available in coffee beans. A tocopherol analysis in coffee beans would be desirable to check this point. So why did authors choose only phytosterols to assess the transfer of bioactive compounds from coffee to coconut oil? Please clarify this issue.

Other comments:

The title is rather confusing, I suggest the authors to modify it to make the aim of the study clearer to the readers.

Writing must be carefully checked throughout the whole manuscript. Just to mention a few examples to be corrected:

Lines 28-29: avoid repetition.

Line 30: “edible oil was low toxicity for humans”.

Line 193: “which may form small molecules of yellow pigment may form when heated”.

Lines 205-206: “the changing pattern of PV slightly different from that of AV”.

I understand that CCO refers to the “coconut coffee oil”, but authors mention CCO regarding coconut oil in lines 9 and 92. Please modify.

Line 41: Schizochytrium limacinum must be written in italics.

In lines 77-83, authors mention that their research group already assayed ultrasound treatments with coconut oil and coffee beans and that optimal ultrasound processing conditions were defined. This work must be cited in the manuscript. In the current study, authors use fixed variables (coconut oil:coffee beans ratio, temperature…) and I understand that their values were optimized previously, so that reference is needed to understand why they selected the 5:8 w/w ratio and 50 °C.

Line 92: are coffee beans grinded or not before being added to the coconut oil? Wouldn´t be the transference of bioactive compounds to the oil matrix facilitated if beans are previously grinded? This information should be added to the manuscript. If coffee beans were not grinded, please explain why.

Line 111: mention which FAME standards were used with identification purposes.

Lines 123-125: “phytosterol components were separated and identified by GC-MS on a Hewlett-Packard 7890B/5977A system (Agilent Co., USA) with a SE-124 45 capillary column (50 m × 0.25 mm × 0.1 μm) and a FID”. If compounds were identified by GC-MS, FID should be removed in this sentence.

Lines 125-126: “phytosterols were identified by their retention times and MS sprctra”. Which phytosterol standards were used? Please specify.

Line 165: which alkanes were used for calibration in the analysis of volatile compounds by GC-MS?

Lines 192-194: authors say that yellow color of CCO after thermal ultrasound treatment can be due to carbohydrates and chlorogenic acid which may form small molecules. Considering that coffee beans are a source of carotenoids (as it has been reported in previous works), the yellow color couldn´t be due to the transfer of carotenoids from the coffee beans to the oil? I think that a carotenoid analysis of CCO would have been useful in this work.

Line 199: “AV represents the degree of oil rancidity”. This sentence is not correct because free fatty acids are not responsible of the rancid odors in the oil, but volatile secondary degradation products.

Lines 435-436: “The increase of phytosterol content in CCOs was the main reason for the improvement of antioxidant capacity and thermal oxidation stability of CCOs”. But authors did not measure phenols or carotenoids, which could contribute also to the oxidative stability of CCOs.

The manuscript should be carefully checked as several mistakes regarding language use were detected.

Author Response

(The authors gave the same response as above.)

Reviewer 3 Report

In the present work, the authors investigated the treatment of coconut oil using thermosonic treatment in combination with green coffee beans. The paper is well written and contains a lot of relevant information that will certainly contribute to the scientific development of this area of knowledge. However, here are some suggestions for improvements.

1 – Section 2.4 “Quality indices” – A brief description of the methods should be added. Only the citation of the methods makes the text difficult to read.

Author Response

(The authors gave the same response as above.)

Reviewer 4 Report

The authors did a good work from an experimental point of view, and I recommend the article for publication after some major revisions.

More specific:

L5: Not need for ‘’1’’ if all authors have the same affiliation.

L96: Too much sonication time! All this time you fixed the temperature at 50°C?

L100: Whose color did you measure? The extracted oil, the coconut oil, or the green coffee beans?

L103: The CCO sample was unprocessed coconut oil (L92). Did you analyze only this or extracted oils? You should change the code. CO for coconut oil and the others is right.

L108: When citing ‘’by’’, you should cite for example the first author et al [##].

L124: Is it not the same equipment as L109? Do you mention the company Agilent and Hewlett-Packard model?? Are you sure?

L126: spectra… fix it.

L138: In the DPPH / ABTS radical scavenging assay state the standard you are working on and express the results as mg epigallocatechin gallate (EGCG) equivalent/g oil. But in the abstract (L16) report the results as mg Catechin/g! Which is the right one?

L149: …heated to 500 °C at 10 K/min. Why at Kelvin?

L157: It’s Rancimat! Replace it.

L161: Wrong mention ref. Check again the instructions for authors.

L196: Explain what a,b,c,d means.

L227: Explain in the table footnote what a,b,c,d mean.

L246: The same as above.

L281: The same as above.

L314: Are there statistically significant differences in the samples?

L329: The same as above.

L471: References are not in journal format. Fix them.

Minor editing of English language required. There are some mistakes.

Author Response

(The authors gave the same response as above.)

Round 2

Reviewer 2 Report

The current title (Ultrasound coupled coconut oil extraction Hainan Robusta coffee bean oil quality evaluation of coffee coconut oil for emphasis was placed on coffee coconut oil) is confusing and rather disorganized yet. I suggest authors to modify the title for readers to get a clear idea of the study´s aim.

Authors state in the manuscript that the yellow color of CCO can be due to small yellow molecules formed by carbohydrates and chlorogenic acid transferred from coffee beans to the oil. As the transfer of chlorogenic acid to the oil is probably low, as the authors also recognize, the possibility that carotenoids can contribute to the yellow color is a viable hypothesis (carotenoids are lipid-soluble molecules and therefore more prone to be mixed with coconut oil). Authors just mention that carotenoids in coffee beans can be related to the CCO color, but I think that this idea deserves a little further discussion in the manuscript (considering coffee beans as sources of carotenoids, potential easy transfer of such lipophilic compounds to the oil, thus contributing to the characteristic yellow color…).

Why authors selected sterols to assess their content in CCO after the sonication treatments and not other bioactive compounds which are known to be available in coffee beans is not clearly stated in the manuscript. It is right that authors selected sterols because of their abundance in coffee beans and their lipophilic character, as they explained in response to my previous comment, but please add it briefly in the manuscript so readers are clear about it.

Authors explained in response to my previous comment that coffee beans were not grinded when put together with coconut oil so they can be separated after the sonication treatment and then roasted for comercial purposes. It is a reasonable explanation but please add it to the manuscript. The extraction of bioactive compounds from uncrushed beans appears to be less efficient than from crushed beans, so readers might wonder why beans were not crushed to facilitate the transfer of bioactive compounds to the oil.  

Minor comments:

Line 292: "As shown in Fig. 2a&3b", shouldn´t be Fig. 2a & 2b?

Line 357-358: “However, the antioxidant mechanisms of phytosterols remain completely explored”. Is that what authors wanted to say, or that those mechanisms remain not yet completely explored instead?

English language has been improved but some mistakes remains in the text. For instance, in line 471: “Ultrasound treatment can greatly improved the antioxidant capacity and oxidation stability of coffee coconut oil”

Author Response

Thank Reviewer for your suggestions. Please see the attachment.

Reviewer 4 Report

The paper has been revised according to the suggestions and criticisms of the reviewer. In this revised version, the paper has improved its quality but need some minor revisions.

I have some corrections:

L18: ABTS clearance equivalents increased from 0.00±0.00 mg EGCG/g to 45.38±0.87 mg EGCG/g. Rephrase the sentence.

L113: No italics in the text.

L124: Ethanol instead of ethyl alcohol.

L530: Revise the references' format according to the authors' instructions.

Minor editing of English language required.

Author Response

(The authors gave the same response as above.)
